# New Clinical and Immunofluoresence Data of Collagen VI-Related Myopathy: A Single Center Cohort of 69 Patients

**DOI:** 10.3390/ijms241512474

**Published:** 2023-08-05

**Authors:** Luciano Merlini, Patrizia Sabatelli, Francesca Gualandi, Edoardo Redivo, Alberto Di Martino, Cesare Faldini

**Affiliations:** 1Department of Biomedical and Neuromotor Sciences, University of Bologna, 40126 Bologna, Italy; albertocorrado.dimartino@ior.it; 2Unit of Bologna, CNR-Institute of Molecular Genetics “Luigi Cavalli Sforza”, 40136 Bologna, Italy; sabatelli@area.bo.cnr.it; 3IRCCS Istituto Ortopedico Rizzoli, 40136 Bologna, Italy; 4Department of Medical Sciences, Unit of Medical Genetics, Università degli Studi di Ferrara, 44100 Ferrara, Italy; gdf@unife.it; 5Department of Statistical Sciences, University of Bologna, 40126 Bologna, Italy; edoardo.redivo@unibo.it; 6I Orthopedic and Traumatology Department, IRCCS Istituto Ortopedico Rizzoli, 40136 Bologna, Italy

**Keywords:** collagen VI related myopathy, collagen type VI, *COL6A1*, *COL6A2*, *COL6A3*, Ullrich congenital muscular dystrophy, Bethlem myopathy, intermediate phenotype, myosclerosis myopathy, muscle strength, contractures, immunofluorescence

## Abstract

Pathogenetic mechanism recognition and proof-of-concept clinical trials were performed in our patients affected by collagen VI-related myopathies. This study, which included 69 patients, aimed to identify innovative clinical data to better design future trials. Among the patients, 33 had Bethlem myopathy (BM), 24 had Ullrich congenital muscular dystrophy (UCMD), 7 had an intermediate phenotype (INTM), and five had myosclerosis myopathy (MM). We obtained data on muscle strength, the degree of contracture, immunofluorescence, and genetics. In our BM group, only one third had a knee extension strength greater than 50% of the predicted value, while only one in ten showed similar retention of elbow flexion. These findings should be considered when recruiting BM patients for future trials. All the MM patients had axial and limb contractures that limited both the flexion and extension ranges of motion, and a limitation in mouth opening. The immunofluorescence analysis of collagen VI in 55 biopsies from 37 patients confirmed the correlation between collagen VI defects and the severity of the clinical phenotype. However, biopsies from the same patient or from patients with the same mutation taken at different times showed a progressive increase in protein expression with age. The new finding of the time-dependent modulation of collagen VI expression should be considered in genetic correction trials.

## 1. Introduction

Collagen VI is an extracellular matrix protein that forms a microfilamentous network in many tissues, including the skeletal muscle, muscular fascia, myotendinous junction, tendon, adipose tissue, and skin. It consists of a total of six alpha chains, Col6α1–α6, that are encoded by separate genes, *COL6A1*-*COL6A6* [1]. The α4 chain is absent in humans. The alpha 5 and 6 chains have a more restricted and differential distribution in human skin and muscle and may substitute for the alpha 3 chain [2]. No mutations in *COL6A5* and *COL6A6* have been reported to date in muscular disorders. Mutations in the *COL6A1*, *COL6A2*, and *COL6A3* genes result in either the absence or malformation of the microfibrils, causing a spectrum of muscle disorders: Bethlem myopathy (BM, MIM 158810), Ullrich congenital muscular dystrophy (UCMD, MIM 254090), and myosclerosis myopathy (MM, MIM 255600). 

BM is a relatively mild, slowly progressive disorder, with variable onset from prenatal to adulthood, characterized by axial and proximal muscle weakness; variable contractures of the neck, fingers, elbow, knee, and ankles; distal laxity; and skin changes. Progression is also variable and some individuals require supportive means for outdoor mobility, while respiratory involvement is rarely so severe as to require ventilatory support [3,4,5,6]. UCMD is a more severe and progressive disorder, with a congenital onset, characterized by the combination of muscle weakness, proximal joint contractures, striking hyperlaxity of distal joints, and skin changes. Some affected children never acquire independent ambulation, while others lose this ability after a few years. All children develop severe respiratory involvement requiring ventilatory support in the first or second decade of life. Most of them, in addition, develop severe progressive scoliosis [6,7,8,9,10]. MM is a childhood-onset myopathy characterized by slender muscles with a firm, “woody” consistency and progressive contractures of all joints, including the jaws, spine, shoulders, elbows, wrists, fingers, hips, ankles, and knees [11,12,13]. In addition, at the 166th ENMC International Workshop on Collagen VI Myopathies, it was agreed to restrict the label of UCMD to patients who have never walked or have lost the ability to walk by the age of 12, the BM label to patients who are able to walk during adulthood, and the intermediate form (INTM) to patients who lose ambulation during their teens [14].

We have contributed to the discovery of the interplay between mitochondrial dysfunction [15] and defective autophagy [16] in the pathogenesis of collagen VI-related myopathies (COL6-RM) and performed the first clinical trials [17,18,19]. The muscles of collagen VI-null mice and humans with BM or UCMD have latent mitochondrial dysfunction accompanied by ultrastructural alterations of mitochondria and the sarcoplasmic reticulum, and spontaneous apoptosis [15,20]. These alterations are related to defective autophagy with an inability to remove dysfunctional mitochondria, causing the persistent opening of the permeability transition pore and a reduction in ATP synthesis and reactive oxygen species generation [15,16]. Plating *Col6a1*^−/−^ cells on collagen VI or cyclosporin A (CsA), an inhibitor of the mitochondrial permeability transition pore, normalized the mitochondrial response to oligomycin, restoring mitochondrial function, and the treatment of *Col6a1^−/−^* mice with CsA rescued the muscles’ ultrastructural defects and markedly decreased the number of apoptotic nuclei in vivo [15]. Likewise, inducing autophagy with a low-protein diet [16], rapamycin, or CsA decreased apoptotic markers and improved the functional parameters of *Col6a1*^−/−^ mice [16].

The translation of these discoveries into clinical trials confirmed the mutual involvement of mitochondrial dysfunction and defective autophagy. An open pilot trial with CsA in five patients affected by UCMD or BM showed that mitochondrial dysfunction and the increased frequency of apoptosis were largely normalized after 1 month of oral CsA administration, which also increased muscle regeneration [17]. Subsequently, a long-term treatment with CsA in six UCMD patients significantly increased the muscle strength in five of six patients, while motor function did not change and respiratory function deteriorated in all [18]. A one-year open pilot trial with a normocaloric low-protein diet regimen in seven adult BM/UCMD patients showed the reactivation of autophagy markers in skeletal muscle and blood leukocytes [19]. The reactivation of autophagy resulted in decreased myofiber apoptosis and increased mitochondrial function, with a benefit in muscle homeostasis [19]. Mitochondrial dysfunction, as well as in collagen VI-related myopathies, has also been documented in animal models of DMD and LGMD [21,22,23], and in DMD patients [23,24]. Non-immunosuppressive cyclosporins, which retain the PTP-desensitizing properties of CsA, such as Debio-05 and NIM811, have been proven effective in various animal models and in BM, UCMD, and DMD patients’ cells [23,24,25,26,27]. Defective autophagy and its correction in animal models of DMD, LMNA, and XLMTM was reviewed in a ENMC workshop [28].

There is now the concrete possibility of treating patients with COL6-RM with drugs that correct mitochondrial dysfunction and/or defective autophagy [29]. For this reason, it is important to deepen the knowledge of the natural history of the various forms of COL6-RMs, including a quantitative measure of muscle strength and contractures. To date, the clinical history of large cohorts has regarded 49, 145, and 117 patients, respectively, from 11 [30], 10 [31], and 6 [32] different centers worldwide, 60 patients from China [33], and 33 UCMD from a questionnaire-based nationwide survey in Japan [34]. Single center studies have instead involved smaller cohorts: 16 Spanish patients with BM [35], 28 Brazilian patients (seven BM and 21 UCMD) [36], 17 COL6-RM patients from the USA [37], and 13 UCMD patients from London [38]. It should be noted that in all these natural history studies, the patients were divided according to their ability to walk or not, but in no case was muscle strength quantitatively assessed [30,31,33,34,35,36,38,39]. Clinical trial outcome measures should consider that the common feature of all muscle diseases is the loss of muscle fibers first, followed by the loss of muscle strength and function [40]. In addition, manual muscle testing and functional scales have been found to take longer to demonstrate a trend than quantitative measures of strength [40,41]. Therefore, any effective treatment in muscular dystrophy, such as corticosteroids in Duchenne dystrophy, is expected to first increase muscle strength and only subsequently improve or prolong motor function [42,43,44,45].

To further define the clinical and pathomorphological characteristics of these disorders, we have studied the natural history of a large cohort of patients with COL6-RM evaluated at the Istituto Ortopedico Rizzoli. Our main goal was to collect new quantitative measures of muscle strength and the degree of contracture, and immunofluorescence data, hitherto not considered, that can be used to better design future clinical data.

## 2. Results

### 2.1. Characteristics of the Study Population Stratified by Phenotype

Among the 93 patients exhibiting clinical, immunohistochemical, and muscle imaging aspects compatible with the diagnosis of COL6-RM, we included 69 patients with genetic confirmation of *COL6A1*, *COL6A2*, or *COL6A3* mutations. The remaining 24 patients were still awaiting genetic confirmation, comprising two families—one with nine (family No. 2 in [46]) and the other with seven affected individuals spanning three generations. Of the 69 patients with a mutation in one of the three COL6 chains, 33 (48%) displayed the clinical phenotype of BM, 24 (35%) exhibited UCMD, seven (10%) displayed INTM, and five (7%) had MM (Table 1). The average duration between the initial and final visits was 8.4 years (SD 10.2 years), while the average duration between the first visit and the follow-up was 13.0 years (SD 11.1 years).

Ten patients with BM (30%) exhibited symptoms at birth, with a varying combination of congenital hip dislocation (CHD) in four, hypotonia in four, club foot in four, torticollis in three, and contractures of the elbow and knee in one. Among the remaining 23 patients, 11 presented during early childhood and two during their teenage years, experiencing difficulties such as toe walking and trouble running, climbing stairs, or rising from the floor. Six patients only noticed these difficulties during early adulthood, while four were asymptomatic at their initial visit. All patients with INTM and UCMD displayed signs at birth. Two of the five MM patients presented contractures at birth, while the other three developed them during childhood.

A total of 63 patients achieved independent ambulation at various stages, while six UCMD patients were never able to walk independently. All BM patients initiated walking within 18 months, except for three cases where autonomous walking was achieved at 42, 36, and 30 months, respectively (Pts 1, 19, and 12). Notably, patient 1 underwent surgery for CHD at 18 months old and received, at the age of 3, when he was still unable to walk, a muscle biopsy that revealed a marked reduction in collagen VI (Pt 3 in [27]). For this reason, he was diagnosed with UCMD at the time. However, he began walking at 42 months and, during the last checkup at 16 years, demonstrated a good and prolonged walking ability, with only difficulty encountered in climbing stairs. Among the UCMD patients, ten took their first steps within 18 months, while the remaining eight did so between 20 and 36 months. Five out of the seven INTM patients commenced walking before 18 months of age. The age at which the first steps were taken was significantly different only between BM and UCMD (*p* = 0.009).

Fifty-nine patients underwent at least one CK determination. Among them, 13 had normal CK levels, 38 (16 BM, five MM, four INTM, and 13 UCMD) had slightly elevated levels (up to five times the upper normal value), while eight BM patients had maximum values ranging from 9 to 28 times the upper limit of normal. Among these BM patients, seven had elevated CK levels when they were under 21 years old, while one had an elevated level at the age of 30.

### 2.2. Degree of Contracture

The relationship between the phenotype and the frequency of contracture grades (normal, mild, moderate, severe) for each joint is illustrated in Figure 1A. A chi-square test of independence was conducted to assess the association between the phenotype and frequency of contracture grades (normal, mild, moderate, severe) for each joint. The analysis revealed a higly significant relationship between these variables accros all tests (Figure 1B). Specifically, a mild degree of mouth opening (MO) limitation was observed in only two patients with BM (6%), while INTM (43%) and UCMD (33%) patients displayed a mild–moderate degree of limitation. Two non-ambulant UCMD patients (8%) exhibited severe MO limitation. Among MM patients, limited MO was already present at the first visit, and it progressed to a severe level in four patients while still ambulatory. Severe limitation of neck flexion (NF) was absent in BM patients but present in individuals with INTM (43%), UCMD (25%), and MM (80%). Contractures of the elbows, fingers, knees, and ankles were prevalent in most patients, with UCMD patients consistently exhibiting more severe contractures compared to BM patients. Furthermore, all MM patients had severe contractures in these joints.

In patients with MM, the range of motion of the elbows (Figure 2A), fingers, and knees was limited both in extension and flexion, further decreasing the functional range of motion in these joints. A similar but less pronounced bi-directional range of motion limitation was present in non-ambulatory UCMD and INTM patients, as well as in some older BM patients (Figure 2B,C).

### 2.3. Quantitative Measure of Muscle Strength

The knee extensor muscle strength (% predicted) was 29.1% ± 18.9% in 48 patients. It was 39.9 ± 20.8 (range 14–85) in 24 BM patients, 33.0 ± 7.0 (range 26–40) in three MM patients, 14.6 ± 6.5 (range 5–21) in five INTM, and 16.6 ± 4.4 (range 11–25) in 16 UCMD (Figure 3A). The elbow flexion muscle strength (% predicted) was 19.8% ± 13.9% in 45 patients. It was 28.0 ± 15.3 (range 11–65) in 22 BM patients, 20.3 ± 8.1 (range 13–29) in three MM, 13.0 ± 5.7 (range 6–19) in five INTM, and 9.97 ± 3.6 (range 4–17) in 15 UCMD patients (Figure 3B). The hand grip muscle strength (% predicted) was 30.2 ± 21.3 in 46 patients. It was 42.9 ± 20.9 (range 9–87) in 25 BM patients, 24.3 ± 11.7 (range 14–37) in three MM patients, 16.4 ± 8.2 (range 8–28) in five INTM patients, and 12.2 ± 3.6 (range 6–18) in 13 UCMD patients (Figure 3C). The muscle strength difference was highly significant between BM and UCMD for all the three strength tests (Figure 3D).

### 2.4. Pulmonary Function

Forty-six patients underwent the determination of %FVC at the final visit. The mean %FVC was 71.6 (SD 17.9) in 22 BM patients, 42.7 (SD 10.3) in four MM patients, 36.1 (SD 17.4) in seven INTM patients, and 33.7 (SD 15.7) in 13 UCMD patients (Figure 4). The BM patients had a significantly higher %FVC than the other three groups. However, only nine of the 22 patients with BM had a normal %FVC value (≥80%), seven had values of 79–60%, while six had values below 60%. All patients in the other three groups except one (Pt 56) had values below 60%.

### 2.5. Need for Ventilation and Loss of Ambulation

#### 2.5.1. Need for Ventilation

Thirteen patients were on NIV at the final visit, while 12 patients initiated NIV during the follow-up period (Table 1). The age at initiation of NIV was 24 and 49 years in two patients with BM, 13 and 33 years in two patients with MM, between 15 and 30 years in three patients with INTM, and between 4 and 14 years in 18 patients with UCMD. The age range in the four non-ventilated patients with INTM was 11–21 years and that in the five non-ventilated UCMD patients was 3–9 years. Kaplan–Meier curves depicting the ventilation-free status by age and COL6-RM phenotype demonstrated a statistically significant (*p* < 0.001) difference between the four COL6-RM groups (Figure 5A). The need for non-invasive ventilation occurred in 50% of patients with UCMDL by 11 years of age and by 30 years of age in INTM patients (Figure 5A).

#### 2.5.2. Loss of Ambulation

At follow-up, 30 patients were able to walk without aid, five were still able to take a few steps indoors with assistance (two INTM and three UCMD), while 28 needed a wheelchair. The mean age at loss of ambulation was 44.6 years (SD 9.7 years) in five BM, 17.0 years (SD 1.8 years) in five INTM, and 8.1 years (SD 2.5 years) in 15 UCMD. Of the three patients with MM, one lost their walking ability at age 11 and the other two at age 48. Kaplan–Meier curves showing the probability of loss of ambulation by age and COL6-RM phenotype demonstrated a statistically significant (*p* < 0.001) difference between the four COL6-RM groups. Loss of ambulation occurred in 50% of patients with UCMD by 8 years of age, by 17 years of age in INTM patients, and by 48 years of age in MM patients (Figure 5B).

### 2.6. Survival

Five patients died due to respiratory problems. Patient 34 declined NIV and died at age 36 years. Patient 53 died at age 10 years, a few months after having initiated NIV. The other three patients (Pts 58, 59, and 60) died between the ages 10 and 14 due to intermittent respiratory problems while on NIV.

### 2.7. Collagen VI Immunofluorescence Analysis in Muscle Biopsy

In COL6-RM biopsies, we identified four primary collagen VI patterns (Table 2 and Figure 6): A—the complete absence of collagen VI; MR—a marked reduction in collagen VI expression within the basal lamina of most muscle fibers and endomysial capillaries, associated with abnormal protein accumulation in endomysial regions; mR—a moderate reduction in collagen VI within the basal lamina of some muscle fibers; and N—indistinguishable from normal controls, displaying continuous labeling of muscle fibers and capillary vessel basal lamina.

Muscle biopsies from 13 out of 15 BM patients with dominant mutations exhibited a normal collagen VI pattern, as evidenced by the continuous overlap with the anti-perlecan staining (Table 2). In the case of patient 33, a muscle biopsy taken at the age of 3 displayed a marked reduction in collagen VI within the basal lamina of muscle fibers and capillary vessels, consistent with the MR pattern (Table 2). Interestingly, patient 12, who carried the same mutation as patient 33, demonstrated a moderate reduction (mR pattern) in collagen VI expression during the initial biopsy, at 11 years of age (Figure 7A). Surprisingly, upon re-evaluating collagen VI in a biopsy taken 18 years later from the same patient 12, a normal pattern indistinguishable from the control group was observed (Figure 7B).

Muscle biopsies from patients carrying recessive BM and MM mutations exhibited a moderate reduction in collagen VI within the basal lamina of muscle fibers (mR pattern). However, an exception was observed in BM patient 1 (Pt 3 in [27]), who was examined at the age of 3 and displayed a more severe deficiency in collagen VI, consistent with the MR pattern (Table 2). Patients 40 and 43, with an INTM phenotype, displayed a marked reduction in collagen VI in the initial biopsies conducted at 3 and 9 years, respectively (Pt 40 in Figure 7A). Interestingly, in the case of patient 40, a second biopsy performed 11 years later revealed a normal amount and localization of protein in the endomysium (Figure 7B).

Previous reports on skeletal muscle biopsies have indicated that patients with UCMD mutations exhibit either a complete absence of collagen VI, as observed in patient 60, or a marked reduction in collagen VI within the basal lamina of muscle fibers, accompanied by aberrant protein accumulation in the endomysium and perimysium (MR pattern). Notably, patient 52 exhibited notable differences in the collagen VI pattern between the initial biopsy performed at 1 year old, which showed almost no presence of collagen VI (A pattern) (Figure 7A), and the second biopsy conducted 9 years later, surprisingly revealing a partial deficiency in collagen VI (MR pattern) (Figure 7B).

### 2.8. Genetic Data

There were six families: three with BM (Pts 3–5, 17–18, 24–32), one with MM (Pts 34–35), and one with UCMD (Pts 53–54), while two patients with INTM (Pts 40–41) resulted from the occurrence of paternal germline mosaicism [54].

In the BM group, 18 patients were sporadic, 14 from three families had an autosomal dominant inheritance, and one patient (Pt 8) from a consanguineous family had an affected brother. Homozygous (three cases), compound heterozygous (one case), and heterozygous (29 cases) mutations were found in *COL6A1* (12 cases), *COL6A2* (12 cases), and *COL6A3* (nine cases).

In the INTM group, four patients were sporadic, one (Pt 45) had a parent affected, and two were half-sisters (paternal germline mosaicism). Heterozygous (six cases) and compound heterozygous (one case) mutations were detected in *COL6A1* (four patients), *COL6A2* (two patients), and *COL6A3* (one patient).

In the UCMD group, 22 patients were sporadic and two with autosomal recessive inheritance were brothers. Homozygous (five cases), compound heterozygous (three cases), and de novo heterozygous mutations (16 cases) were found in *COL6A1* (12 patients), *COL6A2* (ten patients), and *COL6A3* (two patients).

The myosclerosis group consisted of five patients, all with an autosomal recessive inheritance and mutations in *COL6A2*. The homozygous founding nonsense mutation p.Gln819Ter (c.2455C>T) was present in a consanguineous family with two affected (Pts 34 and 35) [13] and in patient 36 whose parents were third cousins from the same restricted area of Central Italy. In the other two patients, the nonsense mutations p.Gln819Ter and p.Arg366Ter (c.1096C>T) were associated with missense changes on the partnering allele (c.2489G>A; p.Arg830Gln in Pt 37 and c.2611G>A; p.Asp871Asn in Pt 38) [53].

Likely pathogenic (LP) variants were present in nine patients (Pts 3–5, 8, 10, 16, 21, 59, 69), while, in another four patients (Pts 37, 43, 51, 56), only the second variant was LP. All the variants of the remaining 56 patients were pathogenic.

Six common variants were detected in 15 unrelated patients. In *COL6A1*, the p.Gly284Arg (c.850G>A) missense change was present in four UCDM patients, the p.Gly293Arg (c.877G>A) in two INTM patients and in one UCDM patient, and the intronic variant c.930+189C>T (causing the insertion of an aberrant exon) in three UCMD patients. In *COL6A3*, two BM patients had the p.Gly2053Val (c.6158G>T) missense mutation, and the c.6210+1G>A splicing mutation was present in one INTM and one UCDM patient. Patient 69, from a nonconsanguineous family of Sephardic Jews, had in homozygosity the nonsense mutation p. Arg468Ter (c.1402C>T) in *COL6A2*, which has been reported to be present in 1.65% of Syrian Jews [57].

## 3. Discussion

In this single center cohort study, we have not only provided clinical and genetic information but also present quantitative data on muscle strength and contractures that can be valuable in clinical trials. Additionally, we have obtained surprising immunohistochemical findings based on repeated muscle biopsies.

For the first time, all four phenotypes are presented in this study. MM has previously been reported only in one consanguineous Italian family with a p.Gln819* (stop codon) homozygous mutation in *COL6A2* [13]. In addition to providing updated information about this family, we now report on three more patients with MM. One of these patients, with distant consanguineous parents, resides in the same small area as the first family and carries the same homozygous mutation. It is noteworthy that, among the 35 patients with BM from the Myology Institute [58], the only patient confined to a wheelchair displayed the early occurrence of contractures in the elbows, fingers, and ankles, as well as a forced vital capacity (FVC) of 60% predicted at the age of 60. This patient carried the p.Gln819* (stop codon) homozygous mutation and hailed from a non-consanguineous Italian family in the same small village as our two families with myosclerosis myopathy. These findings and the other five cases in our series confirm the significant clinical variability of MM patients, both within the same family (Pts 34–35) and with the same mutation (Pts 34–36 and Pt 35 from the Myology Institute [58]). Furthermore, the stratification of the three classic phenotypes based on age and motor function does not apply to MM patients since the loss of ambulation can occur during adolescence or adulthood.

While it has been acknowledged that flexion contractures of the elbows, wrists, ankles, and interphalangeal joints of the last four fingers are characteristic of COL6-related myopathies (COL6-RM), they have not been quantitatively examined until now [30,35,39,58,59]. In this study, we provide documentation of a highly significant correlation between the frequencies of each combination of contracture degree (normal, mild, moderate, severe) and phenotype (BM, INTM, UCMD, MM) for each joint. Specifically, BM patients exhibited fewer moderate/severe contractures in all joints compared to INTM and UCMD patients. Severe contractures affecting all joints were already observed in the initial MM family [13] and have now been identified in three additional patients. Furthermore, patients affected by MM, even during their initial visit while still ambulatory, demonstrated early limitation in mouth opening, which tended to worsen with age. We also observed that joint motion restriction was often not limited to extension but also occurred in flexion. For instance, knee flexion resulting from hamstring contracture is frequently accompanied by knee extension due to simultaneous quadriceps contracture. This phenomenon is observed in some older BM patients and more common in wheelchair-bound UCMD patients [38] but is early-onset and progressive in patients with myosclerosis.

Muscle strength, measured for the first time in a large cohort of patients with COL6-RM, has revealed several unexpected findings. Significant differences in knee extension and hand grip muscle strength have been observed between BM and UCMD/INTM. Furthermore, we have documented the extent of muscle strength loss across different phenotypes. The mean percentage of predicted knee extension muscle strength was less than 40% in BM and less than 18% in INTM and UCMD. The mean percentage of predicted elbow flexion muscle strength was less than 30% in BM and less than 15% in INTM and UCMD. Finally, the mean percentage of predicted hand grip muscle strength was less than 43% in BM and 17% in INTM and UCMD. Notably, we discovered that the biceps muscle exhibited the weakest strength across all phenotypes compared to the normal population, a previously unrecognized finding. This pronounced weakness of the biceps suggests that it is less likely to demonstrate strength gains with treatment in a clinical trial [45]. On the other hand, the quadriceps and finger flexors are preferable options, as they exhibited approximately 30–35% more strength than the biceps in the same BM patients, indicating less fibrosis and fiber loss in these muscles.

The loss of independent ambulation occurred within the expected range in our UCMD and INTM patients [38,60,61]. However, among the BM patients, five out of 33 (15%), and among the MM patients, three out of five (60%), were reliant on a wheelchair. It is worth noting that, prior to the discovery of the gene, only three out 122 reported cases of BM were documented as wheelchair users [62]. Among the 93 most recently reported cases of BM [30,31,32,34], all could walk except for two individuals [35]. Among the 24 Dutch cases, only one was completely wheelchair-bound, while over two thirds of the 15 patients aged 50 years and older preferred using a wheelchair for at least part of their ambulation [62].

Initially, respiratory involvement was considered to be infrequent in BM [63]. However, it has since been recognized that respiratory failure is indeed part of the clinical spectrum and can even occur in ambulatory patients [31,64]. Our study confirms the highly significant relationship between FVC and the three common COL6-RM phenotypes well documented in a previous study [31]. Although our BM patients exhibited a significantly higher %FVC compared to the other three groups, 45% of them had an %FVC below 70% of the predicted value. Two of these BM patients, with an %FVC of less than 50%, initiated respiratory support while still ambulatory. In contrast, all patients in the other three groups, except for one, had %FVC values below 60%, and, during follow-up, 23 out of 36 of these patients required ventilation. The mean age at which noninvasive ventilation was initiated was 10.9 years (±2.3) in UCMD and 21.0 years (±7.9) in INTM. Thus, respiratory compromise, as measured by %FVC and the age at which ventilation was initiated, along with measures of muscle strength and retractions, significantly differed among the three phenotypes that were stratified based on age and motor function.

The analysis of collagen VI in 55 biopsies from 37 COL6-RM patients confirmed previously reported data, highlighting the correlation between collagen VI defects and the severity of the clinical phenotype. However, our analysis of repeat biopsies from the same patients revealed a new and potentially favorable aspect. In these cases, the analysis of biopsies taken at different ages demonstrated striking differences in the collagen VI pattern, indicating the time-dependent modulation of collagen VI expression that had not been reported before. Specifically, we observed a progressive increase in protein expression with age, as evidenced by a clear increment in collagen between the first biopsy (performed on average at 4.6 years of age) and the second biopsy (performed after an average interval of 11 years) in five patients. We currently do not have an explanation for this phenomenon. It is possible that collagen VI mutations may affect mechanisms involved in protein turnover. Autophagy, which is defective in COL6-RM, is involved in intracellular collagen degradation. Mice deficient in autophagic protein Beclin1 exhibited a profibrotic phenotype, with increased collagen deposition [65]. Capillary morphogenesis gene 2 (CMG2) is a transmembrane surface protein also expressed by muscle cells, which mediates the intracellular degradation of collagen VI. An impairment in the CMG2 activity has been documented in COL6-RM skin fibroblasts [66]. It is noteworthy that patients carrying *CMG2* mutations accumulate abnormal amounts of collagen VI, leading to nodule formation [67]. In addition, muscle interstitial fibrosis is a common change in COL6-RM biopsies, and fibrosis-related factors, such as transforming growth factor β1, promotes the expression of collagen VI chains [68]. In conclusion, this new finding of the time-dependent modulation of collagen VI expression has to be taken into consideration in trials that aim at the genetic correction of mutations to be validated by immunohistochemical analysis of the protein in the muscle.

Interestingly, the original authors who described UCMD have previously noted improvements in the motility or the absence of recognizable disease progression [7,69,70]. A similar stable course over extended periods has also been reported in a BM family [51,71].

## 4. Materials and Methods

### 4.1. Patients and Phenotypes Included in the Study

We studied 69 patients with clinical and genetically confirmed diagnoses of COL6-RM, who attended the Istituto Ortopedico Rizzoli and CNR of Bologna, from 1978 to 2022. The study was approved by the Ethics Committee of the Istituto Ortopedico Rizzoli, Bologna.

The clinical phenotype was subdivided according to the proposal that we made at the 166th ENMC International Workshop on Collagen VI Myopathies, restricting the label of UCMD to patients who have never walked or have lost the ability to walk by the age of 12, the BM label to patients who are able to walk during adulthood, and the intermediate form (INTM) to patients who lose ambulation during their teens [14]. In addition, we included five patients from four families affected by myosclerosis myopathy (MM), a COL6-disorder allelic to UCMD and BM, which, at the time of the 166th ENMC workshop, was recognized in only one family [13,14].

### 4.2. Data Collection

The following variables were collected: period of onset, creatine kinase (CK) values, contracture’s distribution and grade of severity, presence of congenital hip dislocation, age at first steps and age at loss of ambulation, quantitative muscle strength, forced vital capacity, and age at initiation of noninvasive ventilation (NIV). After the last visit, we obtained the most recent information about the loss of ambulation, need for ventilation, and death by telephone or email from the patients or their parents.

### 4.3. Evaluation of Contractures

The passive range of motion was measured with a standard two-arm goniometer and joint contractures were graded as follows: absent (0), moderate (1), marked (2), severe (3). For elbow flexion, a moderate contracture was 1–29°, marked 30–60°, and severe >60° [72]. For ankle plantar flexion, a moderate contracture was 1–14°, marked 15–30°, and severe >30° [72]. Back flexion contracture is usually accompanied by neck flexion contracture, which is easier to evaluate. For neck flexion, a moderate contracture was 1–14°, marked 15–30°, and severe >30°. The contracture of mandibula movement (mouth opening—MO) was considered absent when the maximum opening distance between the upper and lower central incisors was more than three vertically aligned fingers or more than 51 mm, moderate (1) if 41–50 mm or only three fingers, marked (2) between 36 and 40 mm or less than three fingers, and severe (3) <35 mm or less than two fingers. Knee contracture was evaluated with the patient in the supine position with the thigh maintained 90° of hip flexion and applying pressure on the posterior ankle until the maximum extension was reached and the angle of knee flexion represented the point of hamstring tightness [73], and in the prone position by applying pressure on the anterior ankle until the maximum flexion was reached and the angle of knee extension represented the point of quadriceps tightness. The normal range of knee flexion/extension is 150°: no contracture (0) <10°, moderate (1) 10–44°, marked (2) 45–80°, and severe (3) >80°. Only the highest score of knee contracture was reported for each patient. Finger flexion was evaluated as described [46] and rated as moderate (1) if limited to the interphalangeal joint, marked (2) if the metacarpophalangeal joint was involved, and severe (3) when the wrist was also limited.

### 4.4. Muscle Strength Measures

A handheld dynamometer (CT 3001, Citec, C.I.T. Technics BV, Groningen, The Netherlands) was used to measure the muscle strength of elbow flexion and knee extension in Newtons. Only the highest score obtained on either side at the last visit was used for further analysis and reported as the % of predicted by sex and age [74,75,76]. A Jamar dynamometer (J A Preston Corporation, New York, NY, USA) was used to measure grip strength in kilograms. Hand grip strength was measured in the sitting position with the measurer supporting the weight of the dynamometer by resting it on the palm of the hand. The highest score obtained on either side at the final visit was used for further analysis and reported as the % of predicted by sex and age.

### 4.5. Immunohistochemical Analysis

A diagnostic biopsy was obtained in 37 patients (19 BM, 4 MM, 2 INTM, 12 UCMD). The age of patients at the first biopsy coincided with that of the initial examination. In addition, a second/third biopsy was obtained in 14 patients who underwent surgical treatment for the correction of Achilles tendon contractures, or who participated in the “Cyclosporin A” and “Low-Protein Diet” trials [17,18,19], for a total of 55 biopsies. Collagen VI immunofluorescence analysis was performed on 7-μm-thick frozen muscle sections by double labeling with anti-perlecan antibody as a control protein [77].

### 4.6. Genetic Analysis

Patient genomic DNA was extracted from blood samples by standard procedures. Genetic analysis of the COL6A genes was performed by using either Sanger sequencing or a next-generation sequencing (NGS) protocol based on a custom COL6A gene panel. Pathogenic and likely pathogenic variants (according to ACMG classification (https://www.engenome.com/; accessed on 1 June 2023) were considered.

### 4.7. Statistical Analysis

The statistical analyses were conducted using the programming language R. Graphs were generated using the R package ggplot2. Variables were expressed as frequencies, percentages, and mean ± standard deviation, as appropriate. A chi-squared test of independence was performed to examine the relation between the phenotype and grade of contracture within the 6 different joint contractures examined. The *p*-values are derived from the asymptotic chi-squared distribution; sometimes, the approximation can be poor due to low counts, but the results are equivalent in terms of significance even if the *p*-values are approximated via a Monte Carlo method. For each set of predictions of muscle strength (KE, EF, HG) and FVC, an ANOVA model was fit, having the phenotype as the only factor. A post-hoc analysis was carried out comparing all pairwise means among the phenotypes with the significance given by Tukey’s range test, which adjusted the *p*-value for the multiples. *p*-values < 0.05 were considered statistically significant.

## Figures and Tables

**Figure 1 ijms-24-12474-f001:**
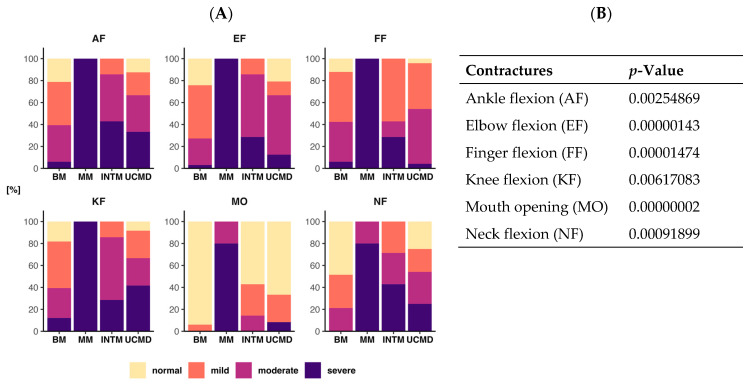
Bar chart showing frequencies for each grade of contracture (normal, mild, moderate, severe) and phenotype (BM, MM, INTM, UCMD) among the six joints tested (AF, EF, FF, KF, MO, NF) (**A**). The chi-square test of independence showed that the relationship between these variables was higly significant for all six different joint contractures examined (**B**).

**Figure 2 ijms-24-12474-f002:**
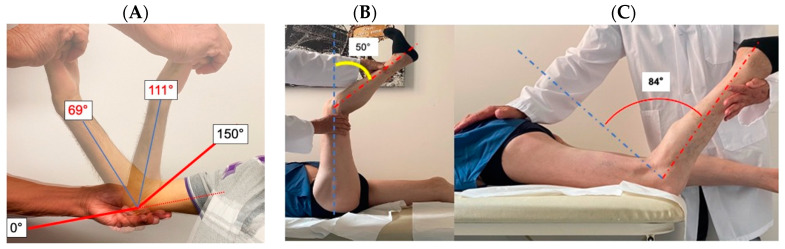
Bi-directional limitation in the range of motion in COL6-RM patients. The normal range of motion of elbow is 0–150° (full extension–full flexion). In this 55-year-old MM patient, the elbow could be moved in extension up to 69° and in flexion up to 111°, for a range of motion of only 42 degrees compared to the normal 150° (**A**). The normal range of knee flexion/extension is 150. In this 67-year-old BM patient, in the supine position, due to the contracture of the hamstrings, the knee could be extended 100° and remained flexed 50° (**B**). In the same patient in the prone position, due to the quadriceps contracture, the knee could only be flexed 66° (knee extension 84°) (**C**). The combined knee extension/flexion movement was 166° (100 + 66) compared to the normal 300° (**B**,**C**).

**Figure 3 ijms-24-12474-f003:**
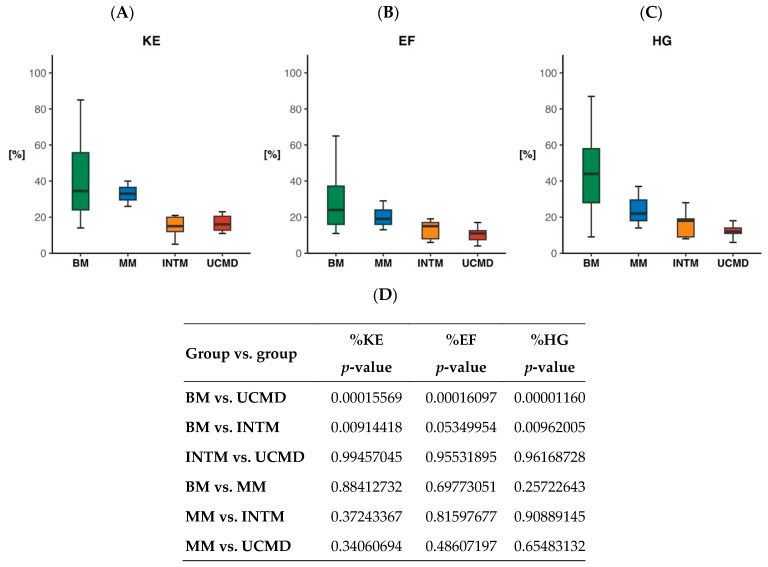
Box plots of percentage of predicted knee extension (**A**), elbow flexion (**B**), and hand grip (**C**) muscle strength in BM, MM, INTM, and UCMD patients. Significance of group versus group comparison (**D**).

**Figure 4 ijms-24-12474-f004:**
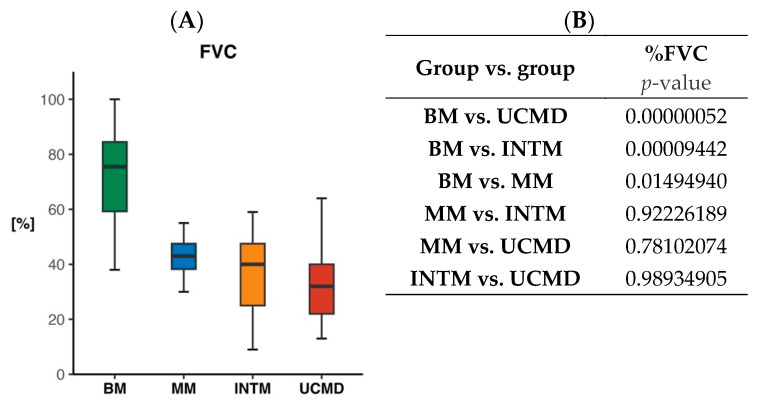
Box plots of percentage of predicted FVC (**A**) in BM, MM, INTM, and UCMD patients. Significance of group versus group comparison (**B**).

**Figure 5 ijms-24-12474-f005:**
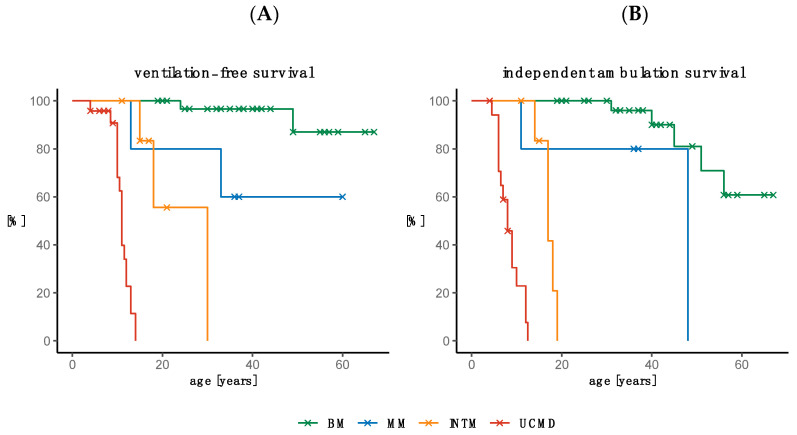
Kaplan–Meier curves depicting ventilation-free status by age and COL6-RM phenotype (**A**). Kaplan–Meier curves showing the probability of loss of ambulation by age and COL6-RM phenotype (**B**).

**Figure 6 ijms-24-12474-f006:**
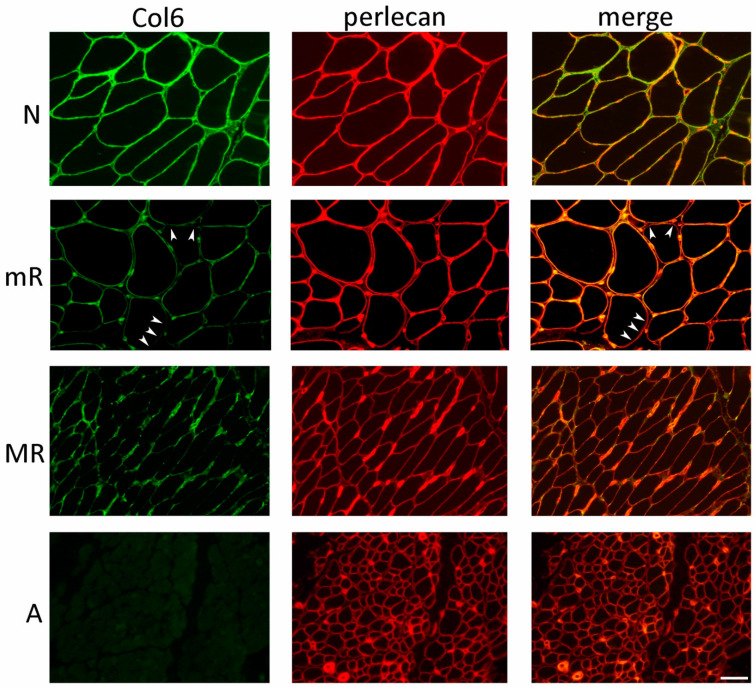
Immunofluorescence analysis of collagen VI (Col6, green) and the basement membrane component perlecan (red), and superimposed images (merge) on cross-sections of muscle biopsies of COL6-RM patients, showing the 4 main immunohistochemical patterns: N, normal expression and localization of collagen VI, as demonstrated by the overlap of collagen VI with the anti-perlecan staining in merge image; mR, partial deficiency in collagen VI in the basement membranes of muscle fibers (arrowheads), also evidenced by partial co-localization with anti-perlecan antibody; MR, diffuse loss of collagen VI labeling in the basement membrane of muscle fibers and accumulation of mutated protein in perivascular endomysial areas; A, absence of collagen VI. Scale bar, 100 µm.

**Figure 7 ijms-24-12474-f007:**
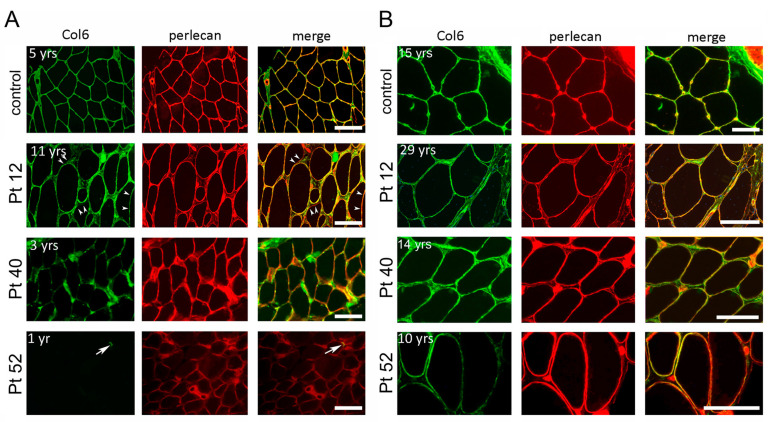
Immunofluorescence analysis of collagen VI (Col6, green) and the basement membrane component perlecan (red), and superimposed images (merge) on cross-sections of muscle biopsies of a healthy control and Col6-RM patients, performed at the indicated age. The first muscle biopsy in the three patients showed a mild reduction in collagen VI in Pt 12 (arrowheads), a marked reduction in Pt 40, and only a trace in patient Pt 52 (arrow). Scale bar, 50 μm (**A**). The repeated biopsies from the same Col6-RM patients show a striking increase in collagen VI amount around muscle fibers, with respect to the first biopsy. The change in the collagen VI pattern is particularly evident when comparing the first biopsy (shown in (**A**)) with the second biopsy (shown in (**B**)) in Pt 40 and Pt 52. Scale bar, 50 µm.

**Table 1 ijms-24-12474-t001:** Characteristics of the study population, stratified by phenotype.

							Contractures										Muscle Strength			
Patient	Phenotype	Onset	Age at First Visit, y	Age at Last Visit, y	Age at FU, y	CK, Times Normal	MO	NF	EF	FF	KF	AF	First Steps, m	Lost Walk, y	NIV, Age, y	%FVC	%KE	%EF	%HG	Mutation
1	BM	B	2	16	20	2	0	2	0	0	0	2	42	-	-	91	41	15	28	H *COL6A2* c.1970-9G>A [47]
2	BM	B	21	35	-	≤1	0	2	1	2	1	3	18	-	24	38	24	13	19	h *COL6A1* c.868G>C
3	BM	As	10	18	26	24	0	0	0	1	1	0	12	-	-	77	85	65	59	h *COL6A3* c.4928T>G [48]
4	BM	B	15	28	32	28	0	1	1	1	1	1	14	-	-	91	85	58	58	h *COL6A3* c.4928T>G [48]
5	BM	As	45	56	-	4	0	0	1	1	1	1	12	-	-	-	-	-	28	h *COL6A3* c.4928T>G [48]
6	BM	B	51	64	65	≤1	0	0	1	2	1	1	12	51	-	69	21	23	87	h *COL6A2* c.875G>C
7	BM	Ch	53	54	57	≤1	0	1	0	2	0	2	12	-	-	-	36	-	41	h *COL6A1* c.1022G>A
8	BM	Ad	47	51	55	≤1	0	1	3	3	1	2	12	45	-	41	16	11	9	H *COL6A2* c.2240T>A
9	BM	Ch	17	29	38	18	1	2	2	2	2	1	12	-	-	79	61	42	43	h *COL6A2* c.2098G>A [19]
10	BM	Ch	17	30	-	3.5	0	2	1	1	2	1	15	-	-	53	31	16	18	ch *COL6A2* c.2947_2952del/chr21 del [49]
11	BM	Ch	20	58	-	1.5	1	2	1	1	2	2	12	56	49	44	24	-	50	h *COL6A2* c.802G>A [48]
12	BM	B	11	35	-	2	0	1	1	2	2	1	36	-	-	53	34	14	36	h *COL6A3* c.6158G>T [48]
13	BM	Ch	41	41	-	UK	0	1	2	2	2	3	12	40	-	58	28	26	17	h *COL6A2* c.1770+1G>A
14	BM	Ch	33	33	-	≤1	0	0	2	1	2	2	12	-	-	80	35	24	46	h COL6A1 c.1056+1G>A [50]
15	BM	B	10	35	-	4	0	1	0	1	2	2	13	-	-	89	61	26	33	H *COL6A3* c.1393C>T [9,48]
16	BM	Ta	29	37	-	13	0	1	1	2	1	2	12	-	-	81	14	48	46	h *COL6A2* c.1970-3C>A [48]
17	BM	Ad	41	41	-	2.2	0	0	1	1	1	0	12	-	-	-	-	-	-	h *COL6A3* c.6166G>T [51]
18	BM	Ch	11	13	25	2.5	0	0	1	1	1	1	16	-	-	85	-	-	-	h *COL6A3* c.6166G>T [51]
19	BM	B	19	25	-	≤1	0	0	1	1	1	2	30	-	-	65	49	22	44	h *COL6A2* c.883G>A [48]
20	BM	B	31	42	-	2	0	2	2	1	2	1	12	-	-	67	24	17	44	h *COL6A3* c.6230G>A [48]
21	BM	Ta	17	17	21	UK	0	0	0	1	0	0	12	-	-	-	-	-	79	c *COL6A2* c.679G>A
22	BM	B	34	40	-	3	0	1	2	2	2	2	12	31	-	63	14	12	12	h *COL6A2* c.847G>A
23	BM	Ad	40	40	-	4	0	2	0	1	0	1	12	-	-	-	-	-	-	h *COL6A2* c.1861G>A
24	BM	Ch	35	67	-	2	0	0	1	1	3	1	12	-	-	93	27	27	56	h *COL6A1* c.428+1G>A [52]
25	BM	Ad	59	59	-	2	0	0	2	3	1	0	12	-	-	-	-	-	-	h *COL6A1* c.428+1G>A [52]
26	BM	Ch	5	33	42	9	0	0	2	2	3	2	12	-	-	-	36	16	26	h *COL6A1* c.428+1G>A [52]
27	BM	Ch	9	32	44	11	0	0	1	2	3	2	12	-	-	-	54	38	-	h *COL6A1* c.428+1G>A [52]
28	BM	As	12	12	38	12	0	0	0	0	0	0	12	-	-	-	-	-	-	h *COL6A1* c.428+1G>A [52]
29	BM	As	17	28	30	5	0	1	1	1	3	0	12	-	-	100	61	35	76	h *COL6A1* c.428+1G>A [52]
30	BM	Ad	14	47	49	15	0	0	1	0	1	1	12	-	-	74	66	44	60	h *COL6A1* c.428+1G>A [52]
31	BM	Ad	65	65	-	UK	0	1	2	2	1	1	12	-	-	-	-	-	-	h *COL6A1* c.428+1G>A [52]
32	BM	Ch	38	61	67	5	0	0	1	2	1	1	12	-	-	83	31	24	58	h *COL6A1* c.428+1G>A [52]
33	BM	B	2.5	2.5	19	3	0	0	0	0	0	0	16	-	-	-	-	-	-	h *COL6A3* c.6158G>T
34	MM	Ch	11	22	36	1.5	2	3	3	3	3	3	15	-	-	41	-	-	-	H *COL6A2* c.2455C>T [13]
35	MM	Ch	16	55	56	2	3	3	3	3	3	3	14	48	33	-	26	13	22	H *COL6A2* c.2455C>T [13]
36	MM	B	6	12	32	1.5	3	3	3	3	3	3	36	11	13 *	30	40	19	14	H *COL6A2* c.2455C>T [13]
37	MM	Ch	32	32	37	1.5	3	3	3	3	3	3	12	-	-	55	-	-	-	ch *COL6A2* c.2455C>T/c.2489G>A [53]
38	MM	B	18	49	60	2.5	3	2	3	3	3	3	14	48	-	45	33	29	37	ch *COL6A2* c.1096 C>T/c.2611 G>A [53]
39	INTM	B	2.5	11	-	2.5	0	1	2	1	2	2	20	-	-	40	12	6	9	h *COL6A1* c.896G>A [54]
40	INTM	B	13	21	-	≤1	2	2	2	1	1	2	17	19	-	47	20	17	28	h *COL6A1* c.896G>A [54]
41	INTM	B	25	25	28	2	1	3	3	3	3	3	18	17	18	21	-	-	-	h *COL6A3* c.6210+1G>A
42	INTM	B	21	21	26	UK	1	3	2	3	3	3	18	14	15	9	-	-	-	h *COL6A2* c.875G>A
43	INTM	B	8	17	-	2.5	0	1	1	1	2	3	15	17	-	59	15	15	19	ch *COL6A2* c.1096 C>T/c.927+5G>A [47]
44	INTM	B	45	45	51	UK	0	3	3	2	2	2	18	18	30	29	5	8	18	h *COL6A1* c.877G>A
45	INTM	B	11	11	15	4	0	2	2	1	2	1	23	-	-	48	21	19	8	h *COL6A1* c.877G>A
46	UCMD	B	5	5	9	3	0	0	0	1	1	0	14	8 *	-	-	12	11	14	h *COL6A1* c.877G>A
47	UCMD	B	5	7	24	1.5	0	1	0	1	0	1	24	10 *	12 *	-	23	11	12	h *COL6A1* c.850G>A
48	UCMD	B	12	20	29	3	0	3	2	1	1	3	20	6	11	13	18	6.5	12	h *COL6A1* c.798_804+8del [47]
49	UCMD	B	5.5	9	22	4	0	1	1	1	2	2	18	6	10*	-	11	14	18	H *COL6A2* c.2626 C>A
50	UCMD	B	3	10	12	1.5	0	2	2	2	2	2	24	6.5	10.5 *	35	17	7	-	h *COL6A1* c.930+189C>T [55]
51	UCMD	B	4	13	14	2	0	1	2	2	2	2	24	12.5	14 *	40	22	8	17	h *COL6A3* c.6210+1G>A [48]
52	UCMD	B	8	12	21	UK	0	2	2	2	2	2	20	8	11	32	18	8	6	ch *COL6A2* c.1459-2A>G/c.1771-1G>A [48]
53	UCMD	B	7	9	10	≤1	0	3	2	2	3	2	18	7	10 *	30	13	12	8	ch *COL6A2* c.1459-2A>G/c.1771-1G>A [48]
54	UCMD	B	5	10	27	≤1	0	1	2	2	3	2	NW	-	13 *	13	13	13	17	h *COL6A1* c.850G>A [27]
55	UCMD	B	5	8	11	≤1	1	3	2	2	3	3	NW	-	4	29	15	5	11	H *COL6A2* c. 2572C>T [48]
56	UCMD	B	6.5	6.5	9	≤1	0	1	0	1	1	2	15	9 *	-	64	14	11	-	ch *COL6A2* c.2098G>A/c.2381C>A
57	UCMD	B	3	11	15	2.5	0	2	2	1	3	3	14	9	12 *	40	20	4	12	h *COL6A1* c.930+189C>T [55]
58	UCMD	B	4	11	14	1.5	3	3	2	2	3	3	NW	-	13 *	-	12	17	8	h *COL6A1* c.819_833del [48]
59	UCMD	B	8.5	8.5	10	1.5	1	3	2	2	3	2	36	6	8.5	45	-	-	-	h *COL6A3* c.6871G>C
60	UCMD	B	1.5	10	11	≤1	1	2	2	2	3	3	NW	-	10	18	-	-	-	H *COL6A1* c.1465del [48,56]
61	UCMD	B	3	3	6	≤1	0	0	1	1	1	1	NW	-	-	-	-	-	-	h *COL6A1* c.904-2A>G
62	UCMD	B	8	8	20	UK	0	0	2	2	2	0	NW	-	14 *	-	22	13	12	h *COL6A2* c.1336dup
63	UCMD	B	3.5	8.5	12.5	3	3	3	3	2	3	3	18	4.5	11 *	22	-	-	-	h *COL6A1* c.930+189C>T; [55]
64	UCMD	B	11	29	36	UK	1	2	3	2	3	3	16	6	11	-	-	-	12	h *COL6A2* c.875G>T
65	UCMD	B	19	19	23	1.5	1	2	3	3	3	3	30	12	10	-	-	-	-	H *COL6A2* c.348dup
66	UCMD	B	10	10	12	UK	1	2	2	1	2	1	14	12 *	11.5 *	57	23	9	-	h *COL6A1* c.850G>A
67	UCMD	B	3	4	8	2	0	0	1	1	1	1	15	-	-	-	12	-	-	h *COL6A1* c.850G>A
68	UCMD	B	1.5	3	4	UK	0	0	0	1	1	1	30	-	-	-	-	-	-	H *COL6A2* c.1402C>T
69	UCMD	B	2	2	7	2	0	0	0	0	0	0	15	-	-	-	-	-	-	h *COL6A2* c.911G>T

FU = follow-up; CK = creatine kinase; MO = mouth opening; NF = neck flexion; EF = elbow flexion; FF = finger flexion; KF = knee flexion; AF = ankle flexion; NIV = non-invasive ventilation; %FVC = force vital capacity, percent predicted; KE = knee extension, percent predicted; EF = elbow flexion, percent predicted; HG = hand grip, percent predicted; y = years; m = months; BM = Bethlem myopathy; MM = myosclerosis myopathy; INTM = intermediate form; UCMD = Ullrich congenital muscular dystrophy; B = birth; As = asymptomatic; Ch = childhood; Ad = adult; Ta = teen age; UK = unknown; NW = never walk; H = homozygous; h = heterozygous; ch = compound heterozygous; * = age at FU.

**Table 2 ijms-24-12474-t002:** Immunofluorescence patterns of collagen VI in the first and repeat muscle biopsies from 37 COL6-RM patients.

Patient	Phenotype	Mutation h, ch, H	I Biopsy Pattern: N, mR, MR, A (Age, Years)	II Biopsy Pattern: N, mR, MR, A (Age, Years)	III Biopsy Pattern: N, mR, MR, A (Age, Years)
4	BM	h	N (16)	N (21)	-
5	BM	h	N (46)	-	-
7	BM	h	N (54)	-	-
9	BM	h	N (16)	N (29)	N (30)
11	BM	h	N (27)	N (48)	N (49)
12	BM	h	mR (11)	N (29)	-
16	BM	h	N (29)	N (36)	N (37)
18	BM	h	N (11)	-	-
19	BM	h	N (23)	N (24)	-
20	BM	h	N (29)	N (41)	N (42)
22	BM	h	N (40)	-	-
26	BM	h	N (15)	-	-
27	BM	h	N (10)	-	-
32	BM	h	N (58)	-	-
33	BM	h	MR (3)	-	-
1	BM	H	MR (1)	-	-
8	BM	H	mR (47)	-	-
10	BM	ch	mR (19)	-	-
15	BM	H	mR (10)	mR (24)	-
34	MM	H	mR (17)	-	-
35	MM	H	mR (31)	-	-
36	MM	H	mR (6)	-	-
38	MM	ch	mR (36)	-	-
40	INTM	h	MR (3)	N (14)	-
43	INTM	ch	MR (9)	-	-
48	UCMD	h	mR (20)	mR (21)	-
50	UCMD	h	MR (3)	-	-
51	UCMD	h	MR (5)	mR (14)	-
52	UCMD	ch	A (1)	MR (10)	-
54	UCMD	h	mR (8)	mR (11)	-
55	UCMD	H	MR (3)	mR (11)	-
57	UCMD	h	MR (3)	-	-
58	UCMD	h	MR (4)	-	-
60	UCMD	H	A (2)	-	-
62	UCMD	ch	MR (3)	-	-
63	UCMD	h	MR (6)	-	-
66	UCMD	h	MR (12)	-	-

h = heterozygous; H = homozygous; ch = compound heterozygous; N = normal; mR = moderate reduction; MR = marked reduction; A = absent; BM = Bethlem myopathy; MM = myosclerosis myopathy; INTM = intermediate; UCMD = Ullrich congenital muscular dystrophy.

## Data Availability

Data are available on request due to privacy or ethical restrictions.

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
