# Peer review of "New Clinical and Immunofluorescence Data of Collagen VI-Related Myopathy: A Single Center Cohort of 69 Patients"

_ijms, 2023, doi:10.3390/ijms241512474_

Round 1

Reviewer 1 Report

The Authors investigate 69 patients (33 BM, 24 UCMD, 7 INTM, 5 MM) performing clinical, immunofluorece and genetic analysis. interesting clinical and immunofluorescent data are reported.

Table1. Mutation of all patients are reported.

The Authors state that the mutations reported in Table 1 are pathogenic or likely pathogrenic.

It is advisable to indicate in the table which are the pathogenic mutations and which
are the likely pathogenic ones
Furthermore, the Authors should indicate the groups of relatives in the Table

 Discussion

Is it possible that collagen VI is expressed in different quantities in different tendons,
have you excluded that the variety of expression is due to this?

Reviewer 2 Report

The manuscript submitted by Merlini and collaborators describes the deep phenotyping data of 69 patients with collagen VI-related myopathies (COL6-RM), aiming to identify innovative clinical data to better design future trials.

They describe and analyze a relevant cohort of patients with the 4 phenotypes of COL6-RM: 33 with Bethlem myopathy, 24 with Ullrich congenital muscular dystrophy, 7 with intermediate phenotype and 5 with myosclerosis myopathy. The last phenotype is not usually included in other cohorts. The strength of the study relies on the detailed description of several aspects of the disease in relation to each specific phenotype, such as the degree and combination of contractures (with relevant data in mouth), the quantitative measure of muscle strength and its importance as an early marker, the evolution of the pulmonary function, and the immunolabeling data and changes overtime in muscle biopsies.

In the introduction, the authors mention several studies (large and small cohorts) that reported the clinical history of patients with COL6-RM. However, at least two relevant studies are missing: Meilleur et al., Neuromuscular Disorders 2015, and Natera-de-Benito at al., Neurology 2021. The study of Meilleur et al. describes a two-year pilot study designed to evaluate the feasibility, reliability, and validity of various outcome measures, particularly the Motor Function Measure 32 in patients with COL6-RD and LAMA2-RD, and among other aspects include some validated motor scales. Merlini et al. emphasize that manual muscle testing and functional scales have been found to take longer to demonstrate a trend than quantitative measures of strength. This is an important point to consider in clinical trials. However, it would have been interesting to have both in the same cohort of patients. The Study of Natera-de-Benito et al., includes 117 patients with COL6-RM and proposes a prospective phenotypic classification for COL6-RDs that will enable an accurate prediction of a patient's COL6-RD phenotype during the first years of life that could help the design of future clinical trials by allowing early stratification of trial cohorts. This is thus a large cohort that should be included in the introduction.

Concerning the assessment of pulmonary function in relation to loss of ambulation, the authors show relevant data for each specific phenotype, but they should compare and discuss their data with other relevant studies such as the one of Foley et al., Brain 2013, mentioned in the introduction (ref 31).

In relation to the collagen VI immunolabeling, the authors show the patterns of decreased expression and mislocalization (intracellular retention?) in different phenotypes. They also mention that in 5 patients, repeated biopsies taken at different ages, demonstrated striking differences in the collagen VI pattern, indicating a time-dependent modulation of collagen VI expression that had not been reported before. They cannot explain the reason. They should explain if the biopsies were taken from the same muscle (as it is well known, muscle imaging shows specific muscle involvement). In this regard, I wonder if the authors think that muscle imaging (analyzed by either MRI or sonography) could be useful data for future trials.

Finally, there is a mistake on page 3: Lines 122 to 130 of the introduction should be deleted. 
